# Advances in Management and Therapeutics of Cutaneous Basal Cell Carcinoma

**DOI:** 10.3390/cancers14153720

**Published:** 2022-07-30

**Authors:** Olivia M. Chen, Keemberly Kim, Chelsea Steele, Kelly M. Wilmas, Nader Aboul-Fettouh, Carrick Burns, Hung Quoc Doan, Sirunya Silapunt, Michael R. Migden

**Affiliations:** 1Department of Dermatology, Division of Internal Medicine, The University of Texas MD Anderson Cancer Center, Houston, TX 77030, USA; oliviamchen.md@gmail.com (O.M.C.); cburns@brunederm.com (C.B.); hqdoan@mdanderson.org (H.Q.D.); 2Department of Dermatology, McGovern Medical School, University of Texas, Houston, TX 77030, USA; keemberly.kim@uth.tmc.edu (K.K.); chelsea.e.steele@uth.tmc.edu (C.S.); kelly.wilmas@ucsf.edu (K.M.W.); naderfettouh@gmail.com (N.A.-F.); sirunya.silapunt@uth.tmc.edu (S.S.); 3Department of Dermatology and Head and Neck Surgery, The University of Texas MD Anderson Cancer Center, Houston, TX 77030, USA

**Keywords:** basal cell carcinoma, basal cell nevus syndrome, immunotherapy, Gorlin syndrome, vismodegib, sonidegib, cemiplimab, Mohs surgery

## Abstract

**Simple Summary:**

Basal cell carcinoma (BCC) is the most common malignancy in humans with a range of treatment options available. Tumor and patient characteristics aid in risk-stratification, which influences treatment considerations. Here, we review the advancements in surgical, topical, field, immunotherapeutic, molecular-targeted, and experimental treatment modalities that can be employed in the correct clinical setting for the treatment of BCC.

**Abstract:**

Basal cell carcinoma (BCC), the most common cancer in humans, is a malignant neoplasm of cells derived from the basal layer of the epidermis. Tumor characteristics such as histologic subtype, primary versus recurrent tumor, anatomic location, size, and patient attributes determine the risk level and acceptable treatment options. Surgical options offer histologic confirmation of tumor clearance. Standard excision provides post-treatment histologic assessment, while Mohs micrographic surgery (MMS) provides complete margin assessment intraoperatively. Additional treatment options may be employed in the correct clinical context. Small and low-risk BCCs, broad field cancerization, locally-advanced disease, metastatic disease, cosmetic concerns, or morbidity with surgical approaches raise consideration of other treatment modalities. We review herein a range of treatment approaches and advances in treatments for BCC, including standard excision, MMS, electrodesiccation and curettage, ablative laser treatment, radiation therapy, targeted molecular therapies, topical therapies, field therapies, immunotherapy, and experimental therapies.

## 1. Introduction

Non-melanoma skin cancers (NMSCs) represent approximately one-third of all malignancies in the United States [1]. Basal cell carcinoma (BCC) incidence worldwide has risen steadily by 3–10% each year with treatment of BCC in the 2013 U.S. Medicare population accounting for over $700 million in healthcare expenditures [2,3].Precise epidemiological data on BCC incidence is limited because NMSC is not reliably recorded by cancer registries. Factors conferring an increased risk of BCC development include ultraviolet radiation exposure, fair complexion, immunosuppression, advanced age, and genetic predisposing conditions (i.e., basal cell nevus syndrome (BCNS), xeroderma pigmentosum) [4]. Prognosis is generally excellent for well-defined, small, slow-growing BCCs in low-risk areas, which can be managed with surgical interventions, destructive localized treatments, and non-destructive topical and field therapies [5]. However, a subset of BCCs demonstrate features that predict an aggressive course (see Appendix A). Some of these features include large size, high-risk histologic subtype (i.e., sclerosing, micronodular, morpheaform, infiltrative, and basosquamous growth patterns), systemic drug resistance, recurrent disease, locally-advanced BCC (laBCC) or metastatic BCC (mBCC). Together, they cause significant morbidity and non-negligible mortality [5,6]. The heterogeneity in behavior of BCCs points to diverse underlying genetic drivers and the need for varied and targeted tools in the treatment paradigm.

Early studies of basal cell nevus syndrome revealed aberrant signaling in the Sonic Hedgehog (Hh) pathway characterized by an inactivating germline mutation in the Hh receptor, Patched (*PTCH1*) [7]. Resultant disinhibition of the transmembrane protein Smoothened (SMO) activates the glioma-associated oncogene (GLI) transcription factors and gene expression governing cell growth and proliferation. Sporadic BCCs also harbor Hh pathway mutations in *PTCH1*, *SMO*, and the negative GLI regulator suppressor of fused (*SUFU*) [7]. A recent genomic analysis of BCCs, revealed Hh pathway mutations in 85% of BCCs; however, the same proportion of BCCs harbored additional mutations in other major cancer-related pathways such as N-Myc (30% of BCCs) and Hippo-YAP (*PTPN14* 23% and *LATS1* 8%) [8]. These pathways underscore the diverse and complex underlying molecular drivers of oncogenesis in BCC and are potential therapeutic targets. Here, we review the advances in the management and therapeutic options for BCC treatment (see Table 1).

## 2. Procedural Treatments

### 2.1. Standard Excision with Postoperative Margin Assessment

The ideal treatment for most BCCs continues to be complete surgical removal [9]. In 1987, Wolf and Zitelli published recommendations on the surgical margins required for the surgical excision of BCCs [10]. Well-circumscribed BCCs smaller than 2-cm in diameter excised with a 4-mm margin had 98% clearance at 5-years. Margins of 2 or 3-mm eliminated 75 to 85% [10,11,12,13,14]. In 2020, Quazi et al. conducted a systematic review to identify necessary surgical margins on a larger sample size [15]. Well-demarcated BCCs smaller than 2-cm excised with 5 mm, 4 mm, 3 mm, and 2 mm margins showed complete excision rates of 94.73%, 92.22%, 90.34%, and 88.15%, respectively. Although the depth of invasion was not quantified, depth of recommended excision is the mid-subcutaneous tissue, or to the fascia, perichondrium, or periosteum if the subcutaneous layer is minimal [10]. For low-risk primary BCCs on the trunk and extremities, (excluding hands, feet, and the pretibial area) with positive postoperative margins, re-excision is indicated [10]. Because margin assessment is postoperative, recommendations for closure include secondary intent or linear closure, rather than tissue rearrangement [16]. Given the high risk of complication following second intention closures and the difficulty in attaining clear margins in larger lesions, each case must be evaluated by a team of specialists [17]. High-risk BCCs, such as the aggressive infiltrative variant which displays an increased recurrence risk, require a more careful approach with accurate and complete evaluation of surgical margins [18]. For more complex, high-risk recurrent tumors, and those on cosmetically sensitive sites, Mohs micrographic surgery is recommended [16].

### 2.2. Mohs Micrographic Surgery

Standard surgical excisions have recommended guidelines on appropriate surgical margins for low-risk BCCs on low-risk anatomic locations [16]. In contrast, higher-risk BCCs in higher-risk anatomic locations are more suitable for Mohs micrographic surgery (MMS), a technique in which tumors are excised with narrow surgical margins and intraoperative, frozen-section histological examination of the entire peripheral and deep margins is performed by the surgeon [16]. Residual tumor identified is excised by the same process, which is repeated until margins are tumor-free. MMS is a tissue sparing technique with lower recurrence rates compared to standard surgical excision [12,19,20]. According to a case series by Frederick Mohs and Anthony Emmett, cancers treated by MMS have recurrence rates around 1–2% with MMS demonstrating the lead cure rate for operable BCCs [19]. Two meta-analyses found the 5-year recurrence rate following MMS for primary and recurrent BCCs to be 1.0% and 5.6%, respectively [12,20]. Comparatively, standard excision 5-year recurrence rates were 10.1% and 17.4% for primary and recurrent BCCs, respectively [12,20]. A prospective study from the Netherlands of high-risk facial BCCs treated with MMS versus standard excision with minimum 10-years follow up revealed recurrence rates of 4.4% after MMS versus 12.2% after standard excision for primary BCCs, although this data did not reach significance (*p* = 0.100). For recurrent BCCs, recurrence rates were 3.9% for MMS and 13.5% for standard excision at 10 years (*p* = 0.023) [21]. Interestingly, 56% of primary BCC recurrences and 14% of recurrent BCC recurrences occurred after 5-years follow up [21]. More recently, a Cochrane Database Systematic Review of RCTs on BCC interventions concluded that MMS provided the lowest recurrence rates for BCC compared to standard excision and other non-surgical treatments [22]. In 2012, the American Academy of Dermatology, the American College of Mohs Surgery, the American Society for Dermatologic Surgery Association, and the American Society for Mohs Surgery developed an appropriate use criteria (AUC) for MMS based on tumor and patient characteristics [23]. These criteria stratify tumors into three groups (appropriate, uncertain, and inappropriate) based on tumor subtype, location, size, host immune status, and primary vs. recurrent tumor [23]. These clinicopathological factors influence the number of stages of Mohs surgery required to treat basal cell carcinoma. The MMS AUC is useful for clinicians in determining which BCCs are amenable to standard surgical excision vs. MMS.

### 2.3. Electrodesiccation and Curettage

Electrodesiccation and curettage (ED&C) is a suitable treatment for many low-risk BCCs [16]. The technique requires local anesthesia, two to three cycles of sharp curettage and electrodesiccation of the tumor to firm dermis [16]. The operator must distinguish by feel during curettage, the firm dermis from friable tumor tissue [16]. In certain anatomic areas, ED&C is criticized for operator-dependent results, higher recurrence rates, and cosmetically unappealing scars [24]. Patients with NMSC treated with ED&C have reported inferior cosmetic outcomes to surgical excision or MMS [25]. Recurrence rates range from 3% to 27% for BCCs treated by ED&C [12,13,24,26,27]. Recurrences occurred more often in tumors treated on the nose, paranasal areas, and forehead [24,28]. Aggressive BCC histologic subtypes, such as infiltrative, desmoplastic, morpheaform, or micronodular, displayed higher recurrence rates following ED&C of 27% at 6.5 years [29]. Size also predicted treatment efficacy, with clearance rates at 5-years of 84% for lesions larger than 2 cm and 98.8% for lesions smaller than 1 cm [30]. Other studies have recommended ED&C for BCCs smaller than 6-mm with high cure-rates [31]. Terminal hair-bearing sites also display higher recurrence rate following ED&C [32,33]. ED&C of BCCs should thus be considered for low-risk BCCs in non-facial, non-terminal hair-bearing, and less cosmetically sensitive areas [16]. Close follow up is recommended after ED&C to monitor the scar for tumor recurrence.

### 2.4. Cryotherapy

Cryotherapy is considered a suitable treatment for low-risk BCCs characterized by superficial growth pattern, small size, and low-risk anatomic locations of the trunk and extremities. In one study, the low-risk superficial BCC (sBCC) subtype demonstrated 1-year cure rates of 95.7% when treated with ED&C, compared to 100% with cryotherapy treatment [34]. A phase III clinical trial comparing photodynamic therapy to cryotherapy for treatment of BCCs showed 1 year recurrence rates of 15% for cryotherapy and 25% for PDT [35]. A randomized controlled trial on BCC treatment modalities showed 5-year recurrence rates of 8.2% with standard excision with 3-mm margins and 19.6% for curettage and two freeze-thaw cycles of cryotherapy [14].

### 2.5. Neodymium-Doped Yttrium Aluminum Garnet (Nd:YAG) Laser Ablation

Laser devices have been adopted recently for treating cutaneous malignancies. A pilot study using the neodymium-doped yttrium aluminum garnet (Nd:YAG) 1064 nm laser (80–120 J/cm^2^, 10 ms, 5-mm spot, 4-mm clinical margin, and no cooling) showed a 92% histologic cure rate at 1 month for non-facial BCCs less than 1.5 cm [36]. An expanded study of 31 tumors treated with 1064 nm Nd:YAG (125–140 J/cm^2^, 7–10 ms, 5-to 6-mm spot, 5-mm clinical margin, and no cooling) showed a 90% cure rate at 1-month histologic assessment [37]. Long term follow-up in these studies was limited and operation of the Nd:YAG laser required knowledge of laser endpoints and requisite settings [36,37]. A more recent study coupling Nd:YAG laser with optical coherence tomography (OCT) to treat 119 BCCs initially and again in 2 months, demonstrated clearance of all tumors [38]. The clinical recurrence rate at 1 year was 1.7% [38]. A large systematic review and meta-analysis of Nd:YAG laser treated BCCs (*n* = 3286) had a recurrence rate of 3.1% at median follow-up of 7.9 years [39].

Scars from Nd:YAG laser treatments were noted as superior to surgical scars and tumor clearance persisted at 9 months [40]. Other laser devices, including the pulsed dye laser (PDL), have been used for BCC treatment but with inferior results to the Nd:YAG laser [39,41]. Laser-assisted drug delivery and antibody-targeted gold nanoparticle laser-assisted BCC treatment are also being studied [41].

### 2.6. Carbon Dioxide Laser Ablation

Higher energy pulsed carbon dioxide (CO_2_) 10,600 nm lasers are another option for treating superficial, multifocal BCCs. In a small cohort of 17 BCCs undergoing pulsed CO_2_ laser treatment (500 mJ, 2-4W, 4-mm margin, 3 mm collimated handpiece, 3 passes), complete tumor clearance by histologic assessment was observed [42]. A retrospective review of 61 sBCCs and nBCCs treated with CO_2_ laser ablation revealed a 97% cure rate at a mean 3.4-years follow-up [43]. However, a systematic review and meta-analysis of CO_2_ laser ablation of 904 BCCs showed a 9.4% recurrence rate at median follow-up of 2.1 years [39]. An obvious caveat in comparing these studies is the variable laser settings and ablation protocols employed.

The primary shortcoming of laser ablation compared to surgical modalities is the lack of post-treatment margin assessment. For low-risk BCC treatment, CO_2_ laser ablation has been coupled with reflectance confocal microscopy (RCM) to guide targeted treatment and to confirm tumor ablation via cellular-level resolution [44,45]. A prospective study of 22 low-risk BCCs treated with a single pass of CO_2_ laser ablation showed RCM-imaged residual tumor in 22.7% cases [45]. Additional passes were administered until imaged tumor clearance was achieved, which was maintained at median follow-up of 28.5 months [45]. Pairing CO_2_ laser ablation with RCM is a novel treatment approach with studies underway to identify optimal ablation parameters and imaging protocols, validated by histology [44].

Fractional ablative CO_2_ laser can also be combined with topical medications, such as 5-fluorouracil (5-FU), to enhance percutaneous drug delivery [41]. When coupled with 5-FU, a maximum depth of drug uptake of 1.5 mm was noted an hour after treatment [41]. Carbon dioxide laser ablation followed by topical 5-FU under occlusion for a week resulted in a 71% histological clearance rate of sBCCs at 4 to 8-weeks, which is comparable to topical 5-fluorouracil monotherapy [41]. Carbon-dioxide laser-assisted drug delivery with other topical medications, such as cisplatin 0.1% solution, is being studied [41]. Carbon-dioxide laser has also been proposed in combination with photodynamic therapy, with overall good results [46].

## 3. Topical, Intralesional, and Field Treatments

### 3.1. Topical 5-Fluorouracil

5-fluorouracil (5-FU) is an antimetabolite that impairs DNA replication by blocking the synthesis of thymidylate. The 5-FU 5% cream twice daily for 3–12 weeks is FDA-approved for sBCC treatment [47]. The National Comprehensive Cancer Network (NCCN) Guidelines recommend topical 5-FU for sBCCs in low-risk locations with low-risk features [16]. Some studies have reported sBCC cure rates as high as 90% with topical 5-FU [48]. However, a large, RCT reported lower and less durable efficacy, with 5-year tumor-free survival probability of 70% (95% CI = 62.9–76.0), compared to imiquimod at 80.5% (95% CI = 74.0–85.6 [49]. For the sBCC treatment, a noninferiority randomized controlled trial showed that tumor-free survival at 3-years for 5-FU was 68.2% (95% CI = 58.1–76.3), deemed noninferior to methyl aminolevulinate photodynamic therapy (MAL-PDT) at 58.0% (95% CI = 47.8–66.9) but inferior to imiquimod at 79.7% (95% CI 71.6–85.7) [50]. Treatment with 5-FU for sBCC in low-risk anatomic locations is well-tolerated with common adverse events being erythema and erosion [16].

### 3.2. Topical Imiquimod

Imiquimod is an immunomodulator that antagonizes tumorigenesis in BCC by innate and adaptive immune activation through toll-like receptor (TLR) 7 and 8 agonism and proinflammatory cytokine signaling [51]. Imiquimod also downregulates GLI expression through protein kinase A (PKA) activation [52,53]. The 5% topical formulation is FDA-approved for sBCCs smaller than 2-cm in diameter on the trunk, neck, or extremities of immunocompetent adults [54].

Imiquimod treatment displays different tumor clearance rates for treating different BCC histological subtypes. A prospective trial showed a 5-year cure rate for sBCC treatment at 85% [55]. A phase III randomized controlled trial treating sBCC and nBCC showed an 84% cure rate at 3-years [56]. A systematic review described tumor clearance rates of 43 to 100% for sBCC (*n* = 1482 tumors), 42 to 100% for nBCC (*n* = 438 tumors) and 56 to 63% for infiltrative BCCs (iBCC; *n* = 43 tumors) [57]. The SINS trial (a multi-center, non-inferiority randomized controlled trial) established imiquimod 5% cream as less effective compared to standard excision for low-risk BCCs, with tumor clearance rates of 84% and 98% at 5-year follow up [58,59]. A recent Cochrane database systematic review of 52 RCTs with 6690 participants with nBCC or sBCC, showed that imiquimod compared to surgical excision resulted in more recurrences at 3 years (16.4% versus 1.6%) and 5 years (17.5% versus 2.3%) [22,58]. Interestingly, the rate of good/excellent cosmetic outcomes for imiquimod compared to surgical excision at 3 years was 60.6% versus 35.6% when assessed by investigators; however, no significant cosmetic differences were noted by trial participants at 6-months or 3-years post treatment [22,58]. Topical imiquimod is an acceptable treatment for low-risk BCCs, with excellent cosmetic outcomes. It is overall well-tolerated with common adverse effects being skin irritation, erythema, and rarely erosion [16].

### 3.3. Topical Hedgehog Inhibitors

Hedgehog pathway inhibitors (HPIs) are desirable therapeutics for their targeted molecular inhibition of aberrantly overactive Hh signaling, a major driver of BCC tumorigenesis [60]. In a double-blind, randomized, vehicle-controlled study of 8 patients with BCNS harboring 27 BCC lesions, sonidegib (LDE225) 0.75% cream compared to vehicle applied daily for 4 days resulted in a complete response in 23%, a partial response in 69%, and no response in 8% of treated BCCs [61]. Up to a 16-fold reduction in GLI1, GLI2, and PTCH2 expression was noted in most lesions in the treatment group [61]. However, clinical efficacy has been notably lower in other studies of topical HPIs. A phase II study of sporadic nBCC and sBCC treated with topical sonidegib was terminated early due to lack of efficacy [62]. A phase I study for a novel topical HPI, CUR61414, also did not show efficacy [63]. Drug formulation, penetrance, and rapid drug metabolism are potential culprits. Topical HPIs are still being researched even though they are not yet clinically available. This is because local drug delivery is expected to avoid substantial adverse effects, which frequently cause patients on systemic HPIs to discontinue treatment. Calienni et al. proposed a nano-drug delivery system for topical vismodegib, though not yet used in humans [64]. A phase II study on patidegib 2% gel daily for 12 weeks revealed greater tumor reduction compared to vehicle gel (*p* = 0.038) (NCT02828111) [65]. Patidegib 2% or 4% gel is being evaluated in several clinical trials for patients with BCNS and in non-syndromic patients with high frequency BCCs with results pending (NCT02762084, NCT02828111) [65,66]. A phase III study on patidegib gel 2% for BCCs in 174 BCNS patients was completed in December 2020, with results pending (NCT03703310) [67].

### 3.4. Histone Deacetylase Inhibitors (Vorinostat, Remetinostat)

Histone deacetylases (HDAC) enhance GLI function in the Hh pathway and are promising targets for BCC therapeutics [68]. The topical pan-histone deacetylase inhibitor (HDACi), remetinostat 1% gel applied three times daily for six weeks, was evaluated in a phase II open trial of 25 patients with 33 BCCs [69]. The overall response rate (ORR), defined as a 30% or greater decrease in tumor diameter at week 8, was 69.7% [69]. Pathologic resolution was seen in 54.8% of tumors, but response varied by histological subtype with 100% of sBCCs, 68.2% of nBCCs, and 66.7% of iBCCs responding [69]. No serious adverse events were reported. The most common side effect was treatment site dermatitis [69].

Vorinostat is another HDACi, FDA-approved for cutaneous T-cell lymphoma treatment [70]. It has been studied in advanced BCCs that circumvent SMO inhibition via GLI activation [71]. Both Vorinostat and NL-103, a synthetic chimeric compound with elements of vorinostat and vismodegib, were able to overcome SMO resistance by concurrent inhibition of HDAC function and Hedgehog signaling [71]. However, vorinostat is associated with significant adverse side effects. In a phase III trial of patients with cutaneous T-cell lymphoma, 41% of patients had grade 3/4 adverse events. In a few patients, death was attributable to treatment [72]. Systemic HDAC inhibitors have a broad side effect profile and are thus reserved for severe and refractory BCC disease only [68].

### 3.5. Radiation Therapy

A range of accepted radiation techniques, characteristics, and depth of therapeutic dose delivery exist. However, only therapeutic photons and electrons are endorsed by the NCCN for definitive radiation therapy (RT). Practice patterns with other modalities, such as brachytherapy and superficial X-rays, differ substantially worldwide [16]. Comparisons of heterogeneous studies on different types of RT for BCCs with different characteristics is therefore challenging, with tumor recurrence rates range widely from 3–15% at 5 years in the examined studies [13,21,73,74,75].

As primary treatment, RT is considered for local low-risk and high-risk BCCs in nonsurgical candidates and in cases when surgery would cause substantial morbidity [76]. A 10-year retrospective cohort study of BCCs treated with primary RT (comprised of electrons 19%, photons <2%, superficial X-rays 60%, or combination electrons and superficial X-rays 20%) exhibited overall tumor control rate of 92% at 2 years. Recurrent BCCs treated had a lower control rate of 86% [76]. A recent systematic review and network meta-analysis of 40 randomized trials and five nonrandomized studies comparing primary BCC treatment showed statistically equivalent tumor recurrence rates for external beam radiation (3.5%), standard excision (3.8%), and MMS (3.8%) [77]. A 10-year single center, retrospective analysis of 712 nodular BCCs (nBCC) and sBCCs treated with superficial X-ray therapy reported recurrence rates at 2 and 5 years of 2% and 4.2%, respectively [78]. Size greater than 2-cm and male sex were associated higher recurrence rates [78].

For adjuvant BCC treatment, NCCN guidelines recommend RT for significant perineural invasion or for unattainable clear margins after MMS or complete circumferential peripheral and deep margin assessment [16]. The American Society for Radiation Oncology (ASTRO) task force conducted a systematic review and recommend RT with curative intent for nonsurgical candidates and for positive margins when further surgery is not feasible or would cause cosmetic and functional morbidity [79]. Tumors with negative margins but high-risk features, such as perineural invasion or invasion of muscle, cartilage, or bone may also be treated with RT [79]. In a study of 89 patients with BCC and perineural invasion receiving adjuvant RT, 91% of patients had relapse-free survival at 5 years [80].

Radiation therapy side effects can include radiation dermatitis, depigmentation, telangiectasias, among other long term sequelae [77,81]. The risk of secondary carcinogenesis for RT treatment of BCC is not known. Patients with heightened sensitivity to radiation (i.e., homozygous ataxia telangiectasia, aplastic anemia), syndromic predisposition to carcinogenesis (i.e., Li Fraumeni, xeroderma pigmentosum, BCNS), or poorly-controlled connective tissue disease (i.e., scleroderma) are generally precluded from RT [79].

With regard to cosmetic outcomes, a meta-analysis of 58 studies and 21,000 patients reported both brachytherapy and MMS as having improved cosmesis over external beam radiation and standard excision; recurrence rates at one year were similar across the modalities [82]. Another RCT of facial BCCs treated surgically or with RT revealed superior cosmetic results as assessed by dermatologists and patients at 4-years post-treatment [83].

### 3.6. Photodynamic Therapy

Photodynamic therapy (PDT) is a selective anti-cancer treatment in which an administered photosensitizer is preferentially taken up by neoplastic cells [84]. Targeted irradiation with near infrared light results in reactive oxygen species production causing cellular apoptosis [84]. PDT using 5-aminolevulinic acid (ALA) and methyl aminolevulinic acid (mALA), which are prodrugs of protoporphyrin IX, is an off-label treatment for sBCC [84]. Although mALA is no longer produced in the US, mALA, and ALA are approved for sBCC and thin nBCC in Europe [84].

ALA and mALA boast similar efficacy in treating sBCC and thin nBCC, less than 2-cm in diameter and less than 2-mm in histologic thickness [85]. Cure rates range from 60–100% for nBCC and sBCC treatment, with higher complete response rates in sBCC, truncal location, absent ulceration, and thickness less than 0.5-mm [86]. A phase III study as well as a systematic review and meta-analysis found PDT superior in cosmetic outcomes, but inferior in disease control compared to standard excision and cryotherapy [35,87]. One meta-analysis of 23 studies found PDT comparable to imiquimod for sBCC treatment efficacy at 1 year follow up [88]. However, a RCT of 601 patients with sBCC reported tumor-free survival at 5-years to be 62.7% with PDT and 80.5% for imiquimod [49]. Despite the lower efficacy compared to other modalities, patients with numerous BCCs or a predisposing syndrome such as BCNS may benefit from PDT field treatments [89].

Currently, PDT typically employs near infrared wavelengths for optimal penetration depth. Intralesional PDT to enhance delivery depth has also been described for treating different histological subtypes of BCCs although efficacy was comparable to conventional modalities [90]. PDT still remains an alternative treatment in nonsurgical candidates or for low-risk BCCs in which cosmesis is a priority [16]. Common adverse reactions such as pain, pruritus, erythema, and edema are usually limited to the treated area [16].

## 4. Systemic Therapies

### 4.1. Vismodegib

Vismodegib (Erivedge Capsule, Genentech Inc., San Francisco, CA, USA) is a small molecule inhibitor of SMO targeting Hh pathway over-activation present in most BCCs [91,92]. In the 2009 Phase I study of patients with laBCC or mBCC on vismodegib, the objective response rate (ORR) was 58% with median duration of response (DOR) of 12.8 months [93]. Following this, pharmacokinetic studies determined that vismodegib 150 mg daily produced the highest plasma concentrations [94]. In 2012, Vismodegib 150 mg daily was FDA-approved for treating laBCC and mBCC following robust evidence of efficacy in the Erivance clinical trials (a multicenter, international, single arm, two-cohort, non-randomized study) which demonstrated an ORR of 30% in patients with mBCC and 43% with laBCC [91]. Within the laBCC cohort, 21% demonstrated a complete response (CR). Median duration of response was 7.6 months in both cohorts. Adverse events (AEs) occurring in more than 30% of patients included muscle cramps (68%), alopecia (63%), and dysgeusia (51%). Nearly all patients experienced at least one adverse event, the majority being grade 2 or lower [91]. At 21-month follow-up, respective ORR was 48% (median DOR 9.5 months) and 33% (median DOR 7.6 months) in the laBCC and mBCC groups [95]. Independent analysis of the Erivance data revealed that 65% of laBCC demonstrated significant improvement, while 11% of the cohort experienced progression on vismodegib [96]. An evaluation of vismodegib prophylaxis in BCNS patients (a double-blind, randomized, phase II study) showed decreased BCC incidence and size compared to placebo [97]. Despite the improvement noted with vismodegib in highly morbid BCC disease, 20% of mBCC patients in the Erivance trial progressed, which underscores drug resistance mechanisms that synergistic treatment regimens may address [91]. Vismodegib as neoadjuvant therapy prior to surgery has also been used. An open-label, single arm trial of large (average 12.6 cm^2^ area) and high-risk, operable BCCs showed a 27% decreased surgical defect area after preoperative vismodegib treatment for 3-6 months [98]. Vismodegib is effective for advanced BCCs, although treatment resistance to HPIs can be seen. Vismodegib coupled with other therapeutics are currently being studied.

### 4.2. Sonidegib

Another small molecule HPI, sonidegib (Odomzo, Novartis Int. AG, Basel, Switzerland) was approved in 2015 for treating laBCC [99]. Sonidegib was identified in 2010, noted for high tissue penetration and oral bioavailability, blood-brain barrier penetration, and potent anti-tumor activity in medulloblastoma allograft models [100,101]. Phase I studies in patients with medulloblastoma or advanced BCC identified the maximal tolerable dose to be 800 mg daily and 250 mg twice daily. The AEs were comparable to vismodegib AEs, except for reversible grade 3/4 Creatine Kinase (CK) elevation noted in 19% of patients when doses exceeded the maximal tolerable dose [102]. Primary efficacy analysis of the BOLT trial, a randomized, double-blind, multicenter, phase II study, evaluated a 800 mg and 200 mg dose of sonidegib, which showed equivalent ORRs of 34% and 36% respectively, in laBCC and mBCC groups at median follow up of 13.9 months [103]. At 42-month follow-up, the 200 mg dose had an ORR of 56.1% in laBCC and 7.7% in mBCC with a respective DOR of 26.1 and 24 months [104]. Sonidegib is an effective treatment for laBCC, although mutations conferring resistance are seen in HPI-treated advanced BCCs. In a small trial of patients with vismodegib-resistant advanced BCCs treated with sonidegib for a median of 6-weeks, over half of the patients had progressive disease [105].

### 4.3. Cemiplimab

Cemiplimab, a fully human monoclonal antibody against the PD1 receptor, is an FDA-approved immune checkpoint inhibitor for treating locally advanced cutaneous squamous cell carcinoma (laSCC) and metastatic cutaneous squamous cell carcinoma (mSCC) not amenable to surgery or radiation therapy [106]. In February 2021, cemiplimab garnered FDA approval for treating locally-advanced and metastatic BCC, thereby providing HPI-refractory patients a second-line treatment option [106,107]. BCCs harbor a high mutational burden and are thus expected to respond well to broad anti-tumor immune activity, elicited by immune checkpoint inhibitors such as cemiplimab. In an open label, multicenter, single arm clinical trial of patients with laBCC and mBCC, ORR was noted in 31% of patients by independent review [106]. Complete response was seen in 6% of patients. Forty-eight percent of patients had grade 3–4 adverse side effects, most commonly hypertension. Thirty-five percent of patients exhibited severe adverse side effects. There were no treatment-related deaths. Despite the side effect profile, cemiplimab demonstrated clinical efficacy as a second-line treatment for advanced BCC, whether locally-advanced or metastatic, particularly in those individuals that demonstrate progression, stable disease, or intolerable side effects on HPIs [106,108].

## 5. Investigational Drugs

### 5.1. Taladegib (LY2940680)

Taladegib (LY2940680) is a competitive SMO antagonist that inhibits the activity of vismodegib-resistant SMO mutants at D473H [109]. In a phase I study, taladegib treating laBCC and mBCC, whether treatment-naïve or refractory to HPIs, demonstrated efficacy with 46.8% of patients exhibiting a complete or partial response at 10.2 months duration of response [110]. Like other HPIs, common adverse events included dysgeusia, fatigue, nausea, and muscle spasms [110].

### 5.2. Patidegib (TAK-441)

Patidegib is a semi-synthetic, topical derivative of cyclopamine and functions by SMO inhibition. A phase I trial showed that patidegib 160 mg daily was well-tolerated [111]. A multi-center, double-blind, randomized, vehicle-controlled phase 2 study demonstrated clinical efficacy and safety of patidegib topical 2% and 4% gel for stage I BCCs in patients with BCNS (NCT02762084) [65]. In a subsequent study, patidegib 4% gel showed higher clinical and molecular response [66]. An additional phase 2 study on patidegib 2% gel for BCC prevention in patients with non-syndromic, high frequency BCCs, is pending results (NCT04155190) [66]. A completed multicenter, randomized, double-blind, vehicle-controlled, Phase 3 study evaluating patidegib 2% gel twice daily for over 1 year for reducing disease burden in 174 BCNS patients is pending results (NCT03703310) [67]. Another open-label extension study is pending results on treatment-related adverse events associated with twice daily application in 107 participants (NCT04308395) [112].

### 5.3. LEQ506

LEQ506 is a potent second-generation SMO antagonist with a pyridazine core evaluated in clinical trials from 2010 to 2015 (NCT01106508) [113,114]. In medulloblastoma allograft models, LEQ506 mediated near complete and sustained GLI1 mRNA inhibition [114]. In depilated murine models it decreased GLI1 expression by 80–90% and *PTCH1* mRNA expression by 60–70% after topical application [115]. Phase I studies showed safety of LEQ506 daily for three weeks in adults with advanced solid tumor, recurrent or refractory medulloblastoma, laBCC or mBCC [113]. The maximal tolerated dose was 400 mg with limiting toxicities including fatigue, dyspnea, and grade 3/4 elevations in CK, AST, ALT, and uric acid [113].

### 5.4. ZSP 1602

ZSP 1602 is an oral second-generation SMO inhibitor evaluated in phase I clinical trials from 2019 to 2021 (NCT03734913) [116]. Part 1 included participants with advanced solid tumors including BCC and medulloblastoma, regardless of SMO or GLI1 expression levels. Part 2 Cohort A included participants with esophagogastric adenocarcinoma with SMO or GLI1 overexpression. Part 2 Cohort B included participants with BCC, small cell lung cancer, neuroendocrine neoplasms, and glioblastoma with SMO or GLI1 overexpression. Results from the trial are pending [116].

### 5.5. CK2 Inhibitors (CX-4945)

Casein kinase 2 (CK-2) is a serine/threonine kinase in Hh pathway activation. It prevents GLI transcription factor degradation and phosphorylates SMO thereby enhancing its activity and cell surface concentration [117,118]. CK-2 inhibition has been shown to reduce GLI1 expression and Hh pathway overactivation in many tumors, including lung cancer, lung cancer cell lines, and medulloblastoma [119,120,121]. A phase I open label trial studying CK-2 inhibitor (CX-4945) to treat SMO inhibitor-resistant laBCC and mBCC is underway (NCT03897036) [122]. CX-4945 inhibition is sufficiently downstream in the Hh pathway, which reduces the likelihood of off-target effects and acquired resistance through downstream mutations.

### 5.6. Itraconazole

Itraconazole is an antifungal medication that exhibits SMO antagonism [123]. Topical and oral itraconazole have decreased BCC tumor size in mouse models [124]. In an open-label, phase II trial of oral itraconazole for BCC treatment, 19 of 29 patients were treated with oral itraconazole 200 mg twice daily for 1 month, or 100 mg twice daily for 2.3 months. Itraconazole markedly decreased cell proliferation by 45% (measure by Ki67 tumor proliferation), Hh pathway activity by 65% (measured by GLI1 mRNA expression), and tumor area by 24% [125]. Four of 8 patients with multiple BCCs achieved partial response and the other 4 had stable disease [125]. A Phase 2, open-label, placebo-controlled trial of topical itraconazole 0.7% gel applied for 1–3 months in BCC treatment did not yield reduction of GLI1 mRNA expression or tumor size [126]. Itraconazole is being studied in combination with other SMO antagonists, specifically vismodegib, sonidegib, and with arsenic trioxide [127].

### 5.7. GLI Antagonists (GANTS)

The terminal step of the Hh pathway results in GLI transcription factor activation, nuclear translocation, and gene expression to govern cell proliferation and the epithelial to mesenchymal transition, a critical regulatory first step in the metastatic process [128,129,130]. GLI antagonists-58 and -61 (GANTS) have been studied in cancer cell lines, and GANT-61 has demonstrated anti-tumor effects on xenograft models of neuroblastoma [131,132,133]. No clinical trials are underway for GANTS in BCC treatment.

### 5.8. Anti VEGFR (AIV001)

Vascular endothelial growth factor (VEGF) stimulates the angiogenesis needed for tumor growth, invasion, and metastasis [134]. VEGF expression under certain conditions is regulated by the Hh pathway [135]. Recent studies also demonstrated direct oncogenic effects of VEGF, which alters keratinocyte survival and proliferation to favor NMSC development [136]. Phase I and II open label, non-randomized trials on AIV001, a prolonged-release multi-kinase inhibitor with anti-angiogenic and antineoplastic properties for treating sBCC and nBCC (NCT04470726) [137].

### 5.9. Anti COX-2 TGFB SiRNA

Transforming growth factor-β (TGF-β) is a group of proteins that mediate cell proliferation, apoptosis, migration and epithelial-to-mesenchymal transition, all of which are critical in cancer progression [138]. In iBCCs, TGF-β has been identified in the peritumoral stroma and has been shown to induce peritumoral fibronectin, which promotes cell migration and adhesion needed for tumor infiltration [139]. Overexpression of cyclooxigenase-2 (COX-2), which facilitates prostaglandin formation from arachidonic acid, predicts increased BCC recurrence risk, angiogenesis, and invasive depth [140,141]. STP705, a silencing RNA (siRNA) that knocks down local expression TGF-β and COX-2, is an intralesional medication in current phase II dose-escalation open-label trials for BCC treatment (NCT04669808) [142].

### 5.10. Anti LAG3 Ab

Lymphocyte activation gene 3 (LAG3), also known as CD223, is an immune checkpoint receptor highly expressed on activated regulatory CD4 and CD8 T-cells, tumor infiltrating lymphocytes, B-cells, natural killer cells, and dendritic cells [143,144]. LAG3 amplifies the immunosuppressive function of T-regulatory cells [144]. LAG3 interacts predominantly with MHC-II, preventing T-cell receptor binding and precluding T-cell activation [143]. PD1 and LAG3 are commonly co-expressed on anergic peripheral T-cells, and simultaneous blockade can reverse T-cell anergy and enhance anti-tumor activity [144,145]. This effect was seen in a randomized, double-blind, phase II-III study assessing clinical efficacy of combined treatment with relatlimab, a LAG-3-blocking antibody, and nivolumab, a PD-1-blocking antibody versus nivolumab monotherapy in advanced melanoma patients [146]. Median progression-free survival (PFS) was significantly longer with relatlimab-nivolumab (10.1 months [95% CI 6.4 to 15.7]) compared to nivolumab monotherapy (4.6 months [95% CI 3.4 to 5.6]) [146]. Greater grade 3 or 4 treatment-related adverse events, occurring in 18.9% of patients in the relatlimab-nivolumab group were observed compared to in 9.7% of patients in the nivolumab monotherapy group [146]. The relatlimab and nivolumab combination compared to nivolumab monotherapy is currently in phase II trials for laBCC and mBCC that has progressed on anti PD(L)-1 monotherapy (NCT03521830) [147].

### 5.11. Intralesional Talimogene Laherparepvec (T-VEC)

Intralesional talimogene laherparepvec (T-VEC) is a genetically-modified herpes simplex virus oncolytic virus therapy directly injected into accessible tumors to enhance the local and systemic anti-tumor immune response. Selective viral replication triggers tumor cell lysis and antigen release, which activates tumor-specific effector T-cells [148]. T-VEC has been used to treat melanoma, primary cutaneous B-cell lymphoma (pCBCL), and Merkel cell carcinoma [149]. T-VEC knock-in of granulocyte macrophage colony stimulating factor (GM-CSF) can enhance local and systemic antitumor activity by stimulating other arms of the immune system [150]. A phase I, open label, single arm, single center study on intralesional T-VEC for locally-advanced SCCs, BCCs, Merkel cell carcinomas, and CTCL is ongoing(NCT03458117) [151]. A phase II trial on T-VEC with nivolumab for patients with refractory NMSC, including BCCs, is also ongoing (NCT02978625) [152].

### 5.12. IL2/TNFa

Intralesional therapies offer direct drug delivery to the tumor and decreased systemic drug exposure and related toxicities [153,154]. Bifikafusp alfa + onfekafusp alfa, (Daromun) consists of recombinant interleukin-2 (IL-2) and tumor necrosis factor alpha (TNF-α) fused to an L19 monoclonal antibody (L19IL2/L19TNF) [155]. IL-2 is critical for T-cell development and function, while TNF-α exhibits cytotoxicity to cancer cells [156,157]. The immunocytokine combination is in a phase II clinical study (NCT04362722) for treating high-risk, laBCC or cSCC [158]. In non-surgical candidates with stage III or IV melanoma, the drug has shown CR in 32 melanoma lesions (28.3%; 21 target and 11 non-target lesions) at week 12 [159]. A CR was seen in 7 of 13 (53.8%) non-injected lesions (4 cutaneous, 3 lymph nodes), which suggests that intralesional treatment elicits a systemic immune response [159]. Intralesional L19IL2/L19TNF had primarily low grade 1 and 2 adverse events such as injection site reactions (72.7% of patients) or low-grade fever (59% of patients) [159]. Based on these results, patients with BCC are hypothesized to respond to intralesional L19IL2/L19TNF therapy.

### 5.13. IFN Gamma Adenovirus

Other intratumoral oncolytic virus therapeutics include ASN-002, a replication-defective adenovirus vector that expresses recombinant human IFN-gamma gene in target cells. This decreases cell proliferation via cell cycle inhibition, apoptosis, suppression of angiogenesis, and indirect immune-mediated anti-tumor effects [160,161,162,163]. Targeted cells express IFN-gamma, thereby circumventing toxicities associated with systemic IFN-gamma therapy. An ongoing phase II open-label trial is examining intralesional ASN-002 with vismodegib for sporadic or BCNS-related BCCs (NCT04416516) [164].

## 6. Discussion

Treatment options for BCC range from surgical, intralesional, topical, field treatments, targeted molecular therapies, to immunotherapy. Therapeutic choice relies on comprehensive evaluation of tumor-specific risk factors, including histologic subtype, size, and anatomic location, with patient factors, such immunosuppression, surgical candidacy, therapy-related morbidity, and patient preference. For low-risk and small BCCs, several different therapeutic approaches may offer comparable results but different side effect profiles, thus a patient’s informed preference, concerns about cosmesis, invasiveness of the treatment approach, and risk of adverse events should influence collaborative discussion of treatment options. For larger BCCs, high-risk histology, high-risk anatomic location, or when surgery would result in high morbidity, systemic and targeted molecular therapies should be considered in concert with multidisciplinary input. Immunotherapeutics and targeted molecular therapies have been coupled with definitive surgical modalities, with improved outcomes. The hedgehog pathway has been identified as a major molecular pathway driving the progression of many BCCs, and advances in systemic and local therapeutics targeting this pathway have broadened treatment options for many patients with highly morbid BCC disease. Promising investigational drugs are emerging that target the diverse molecular drivers behind BCC pathogenesis. Additionally, cemiplimab immunotherapy has emerged as a promising treatment for patients who have failed hedgehog inhibitors and other targeted therapeutics. Together, these new drugs promise to broaden further the treatment options for patients with BCC and permit a more individualized approach to BCC treatment.

## Figures and Tables

**Table 1 cancers-14-03720-t001:** Treatments for consideration by category for BCCs with lower or higher-risk features.

Treatments for Consideration by Category ^1^	BCCs with Lower-Risk Features	BCCs with Higher-Risk Features
**Procedural** **treatments**	▪ Standard Excision with postoperative margin assessment ▪ Electrodesiccation and curettage ▪ Cryotherapy ▪ Neodymium-doped yttrium aluminum garnet (Nd:YAG) laser ablation. ▪ Carbon-dioxide laser ablation	▪ Mohs micrographic surgery ▪ Standard Excision with postoperative margin assessment
**Topical,** **Intralesional, and Field Treatments**	▪ Topical imiquimod ▪ Topical 5-fluorouracil ▪ Topical remetinostat (HDAC inhibitor) ▪ Photodynamic therapy ▪ Radiation Therapy	▪ Radiation therapy
**Systemic** **Treatments**		▪ Vismodegib ▪ Sonidegib ▪ Cemiplimab
**Investigational Treatments ^2^**	▪ Patidegib/TAK-441 (SMO inhibitor, topical) ▪ AIV001/Anti-VEGFR ▪ IL-2/TNF-α (intralesional) ▪ IFN gamma adenovirus/ASN-002 (intralesional)	▪ Taladegib/LY2940680 (SMO inhibitor) ▪ LEQ506 (SMO inhibitor) ▪ ZSP 1602 (SMO inhibitor) ▪ CX-4945 (CK2 Inhibitor) ▪ Itraconazole (SMO inhibitor- combined with vismodegib or sonidegib or arsenic trioxide) ▪ Anti LAG3 Ab ▪ Intralesional talimogene laherparepvec (T-VEC) ▪ Anti COX-2 TGF-β siRNA ▪ IL-2/TNF-α (intralesional) ▪ IFN gamma adenovirus/ASN-002 (intralesional) ▪ Vorinostat (HDAC inhibitor)

^1^ Treatments are listed for consideration within the comprehensive clinicopathologic context of the patient under evaluation for treatment. ^2^ Some treatments discussed in this review are excluded from the table because there is no clinically available medication for BCC.

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
