# Peer review of "Advances in Management and Therapeutics of Cutaneous Basal Cell Carcinoma"

_cancers, 2022, doi:10.3390/cancers14153720_

Round 1

Reviewer 1 Report

Authors should better differentiate between low-risk and high-risk BCCs.

The article is rather lengthy and difficult to read; it would be smart to include some illustrations and streamline the descriptions of less common treatments.

The authors should mention the importance of “Clinicopathological factors influencing the number of stages of Mohs surgery for basal cell carcinoma” in Mohs Micrographic Surgery section.

In “Procedural Treatments” Section the authors should introduce a brief discussion about benefits and drawbacks of using cryotherapy with a separate paragraph, instead of only citing it.

The whole manuscript reads like a bullet point presentation, with one-line paragraphs that are not connected to each other. There should be a logical flow from one paragraph to another.

Discussion: you should also include a table summing up all the procedural treatments and try, especially for low-risk and small BCCs, to identify the best treatment options for individual case  and not just listing individual risk factors.

The authors should choose a simpler and more intuitive language. For example:

Lines 17 and 18: Here we review the advancements in surgical, topical, field, immunotherapeutic, molecular-targeted and executive treatment modalities that can be employed in the correct clinical setting in the treatment of BCC.

Line 39: Change “;” with “.”

Line 44: Change “portend” with “are a sign of”

Line 208: Change “exerts dual” with “has two”

Lines 223 and 224: Topical HPIs are still being researched even though they are not yet clinically available.

This is because local drug delivery is expected to avoid substantial adverse effects, which are frequently the reason why patients on systemic HPIs stop taking their treatment.

Line 337: Following that, pharmacokinetic tests revealed that Vismodegib 150 mg daily produced the highest plasma concentrations.

Lines: 347 and 348: An evaluation of vismodegib prophylaxis in BCNS patients (a double-blind, randomized, phase II study) showed decreased BCC incidence and size of developing BCCs as compared to placebo.

Other suggested changes:

Lines 110 and 297: Change “with regard to” with “regarding”

Line 119: Change “The majority of” with “most”

Line 266: Change “range widely, from” with “range widely from”.

line 166 you should add: "CO2 laser has been also proposed in combination with photodynamic therapy, with overall good results." and cite :doi: 10.1111/dth.12616.

line 75 you should add: "Given the high risk of complication following second intention closures and the difficulty in having clear margins in bigger lesions, each case must be evaluated by a team of specialists." and cite: doi: 10.1002/lsm.23502.

Thank You

Author Response

.

Reviewer 2 Report

The manuscript “Advances in management and therapeutics of cutaneous basal cell carcinoma” is a review article regarding the recent advances in treatment approaches for BCC. The manuscript is well written and easy to read. However, there are some concerns that authors should address before the manuscript could be considered for publication:

1.       Not always the data reported are up on date. For instance, in line 38, “BCC accounted for an estimated $4.3 billion in 2012 healthcare expenditures”. Please report more recent data.

2.       The introduction section should be improved providing more histological information on BCC. Indeed, BCC displays several histological variants, each characterized by different behavior and prognosis. Nodular BCC is the most common type, but also the subtype with more favorable prognosis. On the contrary, infiltrative BCC represents an aggressive neoplastic variant (PMID: 33210737).

3.       Line 75: authors should underline that since infiltrating BCC is a more aggressive variant, displaying a high risk of recurrence, this variant requires a more careful approach with an accurate evaluation of its surgical margins (PMID: 34638427).

Author Response

.

Round 2

Reviewer 1 Report

The paper improved a lot after revisions. It is now eligible to be published.

Author Response

The paper improved a lot after revisions. It is now eligible to be published.

Thank you for the comments and suggestions that helped improve the paper.